# Awareness of and positive attitudes toward HIV pre-exposure prophylaxis among reproductive-aged women in Ghana: A multilevel analysis of the 2022 Demographic and Health Survey

**Florence Gyembuzie Wongnaah**[1]*, **Gilbert Eshun**[2], **Mainprice Akuoko Essuman**[3,4], **Collins Adu**[5], **Richard Gyan Aboagye**[6,7]

**1** Abuakwa North Municipal Health Directorate, Ghana Health Service, Kukurantumi, Eastern Region, Ghana, **2** Seventh Day Adventist Hospital, Agona-Asamang, Ghana, **3** Saint Louis University School of Medicine, Saint Louis, Missouri, United States of America, **4** Department of Medical Laboratory Science, School of Allied Health Sciences, College of Health and Allied Sciences, University of Cape Coast, Cape Coast, Ghana, **5** Centre for Social Research in Health, University of New South Wales, Sydney, New South Wales, Australia, **6** School of Population Health, University of New South Wales, Sydney, New South Wales, Australia, **7** Department of Family and Community Health, Fred N. Binka School of Public Health, University of Health and Allied Sciences, Hohoe, Ghana

* florencewongnaah@yahoo.com

## Abstract

### Background

HIV pre-exposure prophylaxis (PrEP) is an effective biomedical intervention for preventing HIV acquisition. Using nationally representative data from Ghana, this study examined the factors associated with awareness of and positive attitudes toward PrEP among women of reproductive age.

### Methods

The study utilised nationally representative data from the 2022 Ghana Demographic and Health Survey, with a weighted sample of 13,980 women. The outcome variable was awareness of and positive attitudes toward PrEP, assessed based on whether respondents had heard of PrEP and approved of people taking it to prevent acquiring HIV. Data analysis was conducted using Stata 18, employing a multilevel logistic regression model to examine factors associated with awareness of and positive attitudes toward PrEP. Results were reported as adjusted odds ratios (aOR) with 95% confidence intervals (CI).

### Results

Overall, 12.1% (95% CI: 11.1–13.3) of women were aware of PrEP and had positive attitudes toward it. The Ahafo region had the highest proportion of women with awareness of and positive attitudes toward PrEP (21.1% [95% CI: 15.7–27.7]), whereas the Savannah region had the lowest (7.2% [95% CI: 5.3–9.9]). Women with two or more sexual partners had lower odds of being aware of and having positive attitudes toward

**Data availability statement:** The data underlying the results presented in this study are third-party data from the Demographic and Health Surveys (DHS) Program. This study used data from the 2022 Ghana Demographic and Health Survey (GDHS). The data are available upon request through registration with the DHS Program at https://dhsprogram.com/data/ The specific dataset used in this study can be accessed at: https://dhsprogram.com/data/dataset/Ghana_Standard-DHS_2022.cfm?flag=1.

**Funding:** The author(s) received no specific funding for this work.

**Competing interests:** The authors have declared that no competing interests exist.

**Abbreviations:** aOR, Adjusted Odds Ratio; CI, Confidence Interval; DHS, Demographic and Health Survey; STIs, Sexually Transmitted Infections; MEASURE DHS, Monitoring and Evaluation to Assess and Use Results Demographic and Health Surveys; HIV, Human Immunodeficiency Virus; AIDS, Acquired Immunodeficiency Syndrome; SSA, Sub-Saharan Africa; WHO, World Health Organization; PrEP, Pre-Exposure Prophylaxis; FSW, Female Sex Workers; MSM, Men who have Sex with Men; GDHS, Ghana Demographic and Health Survey.

PrEP (aOR = 0.51, 95% CI: 0.28–0.96) than those with no partner. Women aged 25–29 (aOR = 1.75, 95% CI: 1.28–2.40), 30–34 (aOR = 1.61, 95% CI: 1.14–2.27), 35–39 (aOR = 1.69, 95% CI: 1.22–2.32), and 40–44 (aOR = 1.48, 95% CI: 1.03–2.12) had higher odds of awareness of and positive attitudes toward PrEP than women aged 15–19 years. Also, higher odds of awareness of and positive attitudes toward PrEP were observed among women who had heard about other sexually transmitted infections (STIs) (aOR = 1.79, 95% CI: 1.40–2.28), had secondary (aOR = 1.34, 95% CI: 1.06-1.70) and higher education (aOR = 3.27, 95% CI: 2.35–4.55), and listened to radio (aOR = 1.20, 95% CI: 1.01–1.43), compared with those who have not heard of STIs, those with no education, and those who did not listen to the radio, respectively. Additionally, women from Central (aOR = 3.28, 95% CI: 1.76–6.12), Volta (aOR = 3.28, 95% CI: 1.63–6.62), Western North (aOR = 2.59, 95% CI: 1.26–5.29), Ahafo (aOR = 5.02, 95% CI: 2.54–9.91), Bono (aOR = 2.87, 95% CI: 1.37–6.02), Bono East (aOR = 2.87, 95% CI: 1.56–5.26), Oti (aOR = 5.64, 95% CI: 3.06–10.39), Northern (aOR = 2.47, 95% CI: 1.10–5.51), Upper East (aOR = 4.17, 95% CI: 2.05–8.50), and Upper West (aOR = 3.62, 95% CI: 1.85–7.08) regions had higher odds of awareness of and positive attitudes toward PrEP than those in the Western region.

## Conclusion

About 1 in 10 women in Ghana were aware of PrEP and held positive attitudes toward it. The factors associated with awareness of and positive attitudes toward PrEP included women's age, educational level, number of sexual partners, listening to the radio, hearing about other STIs, and geographical regions. Policymakers and healthcare providers should focus on interventions that address regional inequalities and barriers to PrEP awareness and facilitate uptake to strengthen HIV prevention efforts in Ghana.

## Background

Human Immunodeficiency Virus (HIV) remains a significant global health issue, with approximately 40 million people living with the virus and an estimated 630,000 deaths reported globally [1]. Sub-Saharan Africa (SSA) continues to bear the brunt of the epidemic, accounting for 65% of global Acquired Immune Deficiency Syndrome (AIDS)-related deaths [1]. In SSA, adolescent girls and young women aged 15–24 years are disproportionately affected, facing infection rates up to eight times higher than their male counterparts [1]. Globally, women account for nearly half of all HIV cases (around 21 million) [2]. Recent estimates show that the African region remains the most severely affected, with one in every thirty adults living with HIV and accounting for more than two-thirds of all persons living with HIV worldwide [2].

Despite significant progress in HIV prevention and treatment, the epidemic in SSA persists, driven by structural inequities, stigma, and unequal access to healthcare [3]. The UNAIDS 95-95-95 targets aim to end the AIDS epidemic as a public health threat

by 2030, yet progress remains uneven, particularly in resource-limited settings [4]. While some sub-Saharan African countries, including Ghana, have achieved reductions in new HIV infections and AIDS-related deaths, many remain off track to meet global goals due to ongoing social and health system barriers [5,6].

Women's increased vulnerability to HIV is influenced by a combination of biological, behavioural, and structural factors, including limited access to sexual health services, socioeconomic inequality, and cultural norms [7]. Biological factors such as mucosal susceptibility increase their risk, while social challenges like gender-based violence and economic dependence further exacerbate their exposure [7]. These factors also worsen women's vulnerability to HIV-related complications, including the risk of mother-to-child transmission during pregnancy, childbirth, and breastfeeding [8]. Additionally, women face a higher likelihood of gynaecological complications such as cervical disorders linked to HPV infection [9] and pelvic inflammatory diseases, including chlamydia and gonorrhoea, which can damage the fallopian tubes [10].

In Ghana, HIV remains a significant public health concern despite relatively low national prevalence compared to other sub-Saharan African countries [11]. However, key populations such as female sex workers (FSWs), men who have sex with men (MSM), and transgender individuals continue to be disproportionately affected, accounting for 28% of all new HIV cases [12]. To address these challenges, the government has implemented a comprehensive national response in collaboration with the President's Emergency Plan for AIDS Relief (PEPFAR) to provide technical support, ensure a consistent supply of antiretroviral drugs, and deliver targeted prevention services for high-risk groups [12].

Over the years, various HIV prevention strategies have been implemented with varying degrees of success. These strategies include HIV testing, counselling, and educational programmes aimed at raising awareness, encouraging safe practices, and lowering the risk of transmission [13]. Among the most promising and relatively new biomedical interventions is pre-exposure prophylaxis (PrEP), which involves HIV-negative individuals using antiretroviral medications to prevent HIV infection [14]. Several clinical trials have shown the effectiveness of daily tenofovir/emtricitabine as an oral PrEP regimen, leading to its approval by the Food and Drug Administration and endorsement by the Centre for Disease Control and Prevention and the World Health Organization (WHO) for individuals at high risk of contracting HIV [14,15]. Consistent use of PrEP reduces the risk of HIV transmission by more than 90% [16]. Maximising the benefits of PrEP depends on adherence, highlighting the importance of understanding and supporting factors that promote consistent use [14].

Despite its proven efficacy, awareness of PrEP and negative attitudes towards it remain significant barriers to its adoption and acceptance, especially among women in resource-limited settings, such as Ghana, who are particularly vulnerable to HIV [3]. PrEP implementation encounters challenges such as stigma, adherence difficulties, and integration with existing healthcare services, as well as economic and personal factors, including affordability, awareness, perceived HIV risk, and willingness to use it [17,18]. Contextual factors, such as the quality of healthcare infrastructure, healthcare providers' knowledge and attitudes, cultural beliefs, and policy support, also significantly influence PrEP uptake and accessibility. These issues are further worsened by community stigma against PrEP users and logistical obstacles [19,20].

Evidence from SSA indicates that awareness of and positive attitudes toward PrEP among women remain comparatively low. For instance, a multilevel analysis across five countries revealed that only 5.61% of women in Tanzania, 6.43% in Burkina Faso, and 9.56% in Côte d'Ivoire were aware of PrEP and held positive attitudes toward it, though the proportion was higher in Kenya (34.29%) [21]. Likewise, a population-based study in Malawi showed that, although some individuals were aware of PrEP, overall awareness and willingness to use it were limited among sexually active adults [22]. In Ghana, recent studies suggest that, despite relatively high awareness and acceptability of PrEP among key populations (such as MSM and FSWs), actual uptake remains low [23–26]. Among FSWs, only 53.59% expressed willingness to use PrEP, and just 6.39% had ever used it [23]. For MSM living in slum communities, barriers such as limited access to healthcare services and distance to facilities hinder uptake [26]. Evidence from Ghana further indicates a gap between awareness, willingness, and actual uptake of PrEP among MSM [27]. Even when awareness exists, misinformation and fear of side effects remain widespread [28].

Although previous studies have focused on MSM, transgender individuals, and FSWs [23–28], little is known about awareness of and positive attitudes toward PrEP among women of reproductive age in Ghana. This study aims to address

this gap by utilising nationally representative data from the 2022 Ghana Demographic and Health Survey (GDHS) to explore the socio-demographic and structural factors associated with awareness of and positive attitudes toward PrEP among women of reproductive age in Ghana.

## Methods

### Study design and description

This study utilised cross-sectional data from the 2022 GDHS. The GDHS is a nationally representative household survey that provides data on various health-related issues, including the transmission, awareness, and prevention of AIDS and other sexually transmitted infections (STIs) [29]. A two-stage stratified cluster sampling technique was employed to sample respondents for the survey. In the first stage of sampling, clusters from Enumeration Areas (EAs) were randomly selected based on the most recent Population and Housing Census. In the second stage, households were systematically sampled from the clusters and EAs, and individuals within the sampled households were then interviewed [29].

Respondents recruited for the survey gave informed consent after the study's objective was explained to them. Participation was voluntary, and respondents were free to decline participation. Our analysis was based on a weighted sample of 13,980 women of reproductive age (15–49 years) with complete data on the outcome and all exposure variables. Details on pre-testing, methods, sampling design, and staff training can be found in the full GDHS 2022 report [29].

### Study variables

**Outcome variable.**  The outcome variable for this study was awareness of and positive attitudes toward PrEP. Respondents were asked whether they had ever heard of PrEP. Those who responded "yes" were asked a follow-up question regarding whether they approved of people taking PrEP to prevent HIV infection. Response options included: (1) never heard of PrEP, (2) heard of PrEP and approve of people taking it to prevent HIV, (3) heard of PrEP but do not approve of people taking it, and (4) heard of PrEP but are unsure about approving it. For the purpose of this study, a composite binary outcome variable was created. Respondents who indicated that they had heard of PrEP and approved of people taking it to prevent HIV were coded as "1" (aware and having a positive attitude). All other responses were coded as "0" (not aware and/or not having a positive attitude toward PrEP).

**Exposure variables.**  The study included fifteen explanatory variables selected based on their availability in the dataset and the literature [21, 22, 27, 30, 31]. The explanatory variables were categorised into individual-level and contextual (household- and community-level variables) based on the hierarchical structure of the DHS data. The variables include the age of the respondent (15–19, 20–24, 25–29, 30–34, 35–39, 40–44, 45–49), educational level (no education, primary, secondary, higher), marital status (never in union, married, cohabiting, previously married), and religion (Christianity, Islam, traditionalist, no religion/other). We also included employment status (not working, working) and exposure to mass media: watched television (no, yes), listened to the radio (no, yes), read a newspaper (no, yes), and used the internet (no, yes). Others included the number of sexual partners excluding spouse (zero, one, two or more), awareness of other STIs (no, yes), and visited a health facility in the last 12 months (no, yes). These variables were considered individual-level variables. Wealth index (poorest, poorer, middle, richer, richest), place of residence (urban, rural), and region of residence (Western, Central, Greater Accra, Volta, Eastern, Ashanti, Western North, Ahafo, Bono, Bono East, Oti, Northern, Savannah, North East, Upper East, Upper West) were treated as contextual-level variables in the multilevel analysis.

### Statistical analysis

The data was analysed using Stata version 18 (StataCorp, College Station, TX, USA). Descriptive analysis was used to describe the study sample, which was weighted (v005/1,000,000). We used weighted sample frequencies and percentages to present the results of the background characteristics of the respondents. Next, we estimated the proportion of

women who were aware of and had positive attitudes toward PrEP using percentages and confidence intervals (CIs). To visualise regional variations, Geographic Information System software was used to generate a thematic map illustrating the regional prevalence of the outcome variable, based on author-generated data and administrative boundary shapefiles. Subsequently, we conducted a bivariate analysis to examine the distribution of the proportion of women who were aware of and had positive attitudes toward PrEP across the explanatory variables. The results were presented using percentages and CIs. Due to the number of explanatory variables, we conducted a significance test using Stata's 'testparm' command, which was executed after the crude regression analysis. This analytical technique helped identify variables significantly associated with women's awareness of and positive attitudes toward PrEP. All variables with p-values < 0.05 were included in the multilevel regression analysis. We used multilevel mixed-effects regression analysis to examine the factors associated with women's awareness of and positive attitudes toward PrEP, given the hierarchical structure of the GDHS data, in which women are nested within households and communities. Four models were developed in multilevel binary logistic regression analyses to identify individual- and community-level factors associated with awareness of and positive attitudes toward PrEP. The first model (the null model) included only the outcome variable. The second model included individual-level variables, whereas the third model included contextual-level variables. In the fourth model, we added both individual- and contextual-level variables. The results were presented as adjusted odds ratios (aORs) with 95% CIs, and statistical significance was assessed using a p-value < 0.05.

### Ethical consideration

The 2022 GDHS obtained ethical approval from the Ghana Health Service Ethics Review Committee and the Institutional Review Board of ICF International. Informed consent was obtained from all participants prior to data collection. This study used secondary data; therefore, additional ethical clearance was not required. However, permission to access and use the dataset was obtained from the Monitoring and Evaluation to Assess and Use Results Demographic and Health Surveys (MEASURE DHS) program after submission and approval of a research project outlining the study objectives and intended use.

## Results

### Background characteristics of the respondents

A total of 13,980 women aged 15–49 years were included in the analysis. Table 1 presents the background characteristics of the respondents. Nearly one-fifth of the women in the study were aged 20–24 (18.1%), with over half having secondary education (62.0%). Regarding marital status, around 40% were married (38.9%). The majority identified as Christians (78.6%) and were employed (74.8%). In terms of media exposure, most women were exposed to television (63.6%), and nearly half listened to the radio (43.6%). Most of the women (76.3%) reported having no additional partners besides their spouse, while 78.2% had heard of other STIs, and more than half (52.4%) visited a health facility in the past 12 months. Wealth was evenly distributed, with nearly a quarter of the women falling within the richer and richest groups, at 23.4% and 23.4%, respectively. More than half of the women lived in urban areas (58.7%), and about one-fifth of respondents were from the Ashanti region (20.2%) (Table 1).

### Awareness of and positive attitudes toward PrEP among women in Ghana

Table 2 and Fig 1 present the results on awareness of and positive attitudes toward PrEP across the background characteristics of the respondents. The overall proportion of women who were aware of and held positive attitudes toward PrEP was 12.1%. Awareness of and positive attitudes toward PrEP were highest among women aged 25–29 (15.8%) and lowest among those aged 15–19 (7.3%). Women with higher levels of education showed greater awareness of and positive attitudes toward PrEP (23.0%). Married women showed higher levels of awareness of and positive attitudes toward PrEP (13.9%). Media exposure also contributed; women who listened to the radio (13.5%) or used the internet (13.8%) showed greater awareness of and positive attitudes toward PrEP. Women living in urban areas had higher awareness of and

**Table 1. Background characteristics of the respondents.**

| Variable | Weighted sample (n) | Weighted percentage (%) |
|---|---|---|
| **Woman's age (years)** | | |
| 15-19 | 2,442 | 17.5 |
| 20-24 | 2,527 | 18.1 |
| 25-29 | 2,207 | 15.8 |
| 30-34 | 2,108 | 15.1 |
| 35-39 | 1,919 | 13.7 |
| 40-44 | 1,559 | 11.1 |
| 45-49 | 1,217 | 8.7 |
| **Educational level** | | |
| No education | 1,920 | 13.7 |
| Primary | 1,903 | 13.6 |
| Secondary | 8,662 | 62.0 |
| Higher | 1,496 | 10.7 |
| **Marital status** | | |
| Never in union | 4,959 | 35.5 |
| Married | 5,441 | 38.9 |
| Cohabiting | 2,093 | 15.0 |
| Previously married | 1,487 | 10.6 |
| **Religion** | | |
| Christians | 10,988 | 78.6 |
| Muslims | 2,523 | 18.0 |
| Traditionalist | 208 | 1.5 |
| No religion/other | 261 | 1.9 |
| **Current working status** | | |
| Not working | 3,524 | 25.2 |
| Working | 10,456 | 74.8 |
| **Read newspaper or magazine** | | |
| No | 13,460 | 96.3 |
| Yes | 520 | 3.7 |
| **Listen to radio** | | |
| No | 7,886 | 56.4 |
| Yes | 6,094 | 43.6 |
| **Watch television** | | |
| No | 5,095 | 36.4 |
| Yes | 8,885 | 63.6 |
| **Use internet** | | |
| No | 8,423 | 60.3 |
| Yes | 5,557 | 39.7 |
| **Number of partners excluding spouse** | | |
| Zero | 10,669 | 76.3 |
| One | 3,071 | 22.0 |
| Two or more | 240 | 1.7 |
| **Heard about other STIs** | | |
| No | 3,050 | 21.8 |
| Yes | 10,930 | 78.2 |

*(Continued)*

**Table 1.** (Continued)

| Variable | Weighted sample (n) | Weighted percentage (%) |
|---|---|---|
| **Visited health facility last 12 months** | | |
| No | 6,650 | 47.6 |
| Yes | 7,330 | 52.4 |
| **Wealth index** | | |
| Poorest | 1,985 | 14.2 |
| Poorer | 2,461 | 17.6 |
| Middle | 2,996 | 21.4 |
| Richer | 3,273 | 23.4 |
| Richest | 3,265 | 23.4 |
| **Place of residence** | | |
| Urban | 8,206 | 58.7 |
| Rural | 5,774 | 41.3 |
| **Region** | | |
| Western | 919 | 6.6 |
| Central | 1,640 | 11.7 |
| Greater Accra | 2,260 | 16.2 |
| Volta | 692 | 4.9 |
| Eastern | 1,183 | 8.5 |
| Ashanti | 2,830 | 20.2 |
| Western North | 391 | 2.8 |
| Ahafo | 299 | 2.1 |
| Bono | 537 | 3.8 |
| Bono East | 600 | 4.3 |
| Oti | 371 | 2.7 |
| Northern | 787 | 5.6 |
| Savannah | 259 | 1.9 |
| North East | 236 | 1.7 |
| Upper East | 601 | 4.3 |
| Upper West | 375 | 2.7 |

positive attitudes toward PrEP (12.9%). Awareness of and positive attitudes toward PrEP were also higher among women who had heard about other STIs (13.3%) and those who visited a health facility in the last 12 months (14.0%). Regarding wealth, women in the richer (13.1%) and richest (13.4%) quintiles had higher awareness of and positive attitudes toward PrEP. Regional disparities were evident, with women from the Ahafo (21.1%), Volta (18.8%), and Northern (18.8%) regions having the highest levels of awareness of and positive attitudes toward PrEP.

### Factors associated with awareness of and positive attitudes toward PrEP among women in Ghana

Table 3 shows the factors associated with awareness of and positive attitudes toward PrEP among women in Ghana. In Model IV, age was a significant factor, with women aged 25–29 (aOR = 1.75, 95% CI: 1.28–2.39), 30–34 (aOR = 1.61, 95% CI: 1.14–2.27), 35–39 (aOR = 1.69, 95% CI: 1.22–2.32), and 40–44 (aOR = 1.48, 95% CI: 1.03–2.12) having higher odds of awareness of and positive attitudes toward PrEP than those aged 15–19. Women with higher educational levels had higher odds of awareness of and positive attitudes toward PrEP (aOR = 3.27, 95% CI: 2.35–4.55) than those with no education. Women who listened to the radio also had higher odds of being aware of and having positive attitudes toward PrEP (aOR = 1.20, 95%

**Table 2. Distribution of awareness of and positive attitudes toward PrEP across exposure variables.**

| Variables | Awareness of and positive attitudes toward PrEP | p-value |
|---|---|---|
| **Overall proportion** | **12.1% (11.1, 13.3)** | |
| **Woman's age (years)** | | <0.001 |
| 15-19 | 7.3 [6.2, 8.8] | |
| 20-24 | 11.0 [9.5, 12.7] | |
| 25-29 | 15.8 [13.8, 18.0] | |
| 30-34 | 14.5 [12.5, 16.9] | |
| 35-39 | 14.0 [11.8, 16.4] | |
| 40-44 | 11.7 [9.6, 14.1] | |
| 45-49 | 11.1 [9.1, 13.5] | |
| **Educational level** | | <0.001 |
| No education | 10.2 [8.5, 12.2] | |
| Primary | 9.2 [7.7, 10.9] | |
| Secondary | 11.4 [10.1, 12.7] | |
| Higher | 23.0 [19.8, 26.5] | |
| **Marital status** | | 0.004 |
| Never in union | 10.7 [9.4, 12.2] | |
| Married | 13.9 [12.4, 15.5] | |
| Cohabiting | 11.6 [9.8, 13.6] | |
| Previously married | 11.4 [9.1, 14.1] | |
| **Religion** | | 0.019 |
| Christians | 12.0 [10.9, 13.2] | |
| Muslims | 13.3 [10.9, 16.1] | |
| Traditionalist | 6.2 [3.3, 11.5] | |
| No religion | 10.4 [5.2, 19.6] | |
| **Current working status** | | <0.001 |
| Not working | 9.4 [8.2, 10.7] | |
| Working | 13.1 [11.8, 14.4] | |
| **Read newspaper or magazine** | | 0.032 |
| No | 12.1 [11.0, 13.3] | |
| Yes | 12.9 [9.7, 16.9] | |
| **Listen to radio** | | <0.001 |
| No | 11.1 [9.8, 12.4] | |
| Yes | 13.5 [12.1, 15.0] | |
| **Watch television** | | 0.001 |
| No | 11.1 [9.9, 12.6] | |
| Yes | 12.7 [11.4, 14.2] | |
| **Use internet** | | <0.001 |
| No | 11.0 [10.0, 12.2] | |
| Yes | 13.8 [12.2, 15.6] | |
| **Number of partners excluding spouse** | | 0.040 |
| Zero | 12.3 [11.2, 13.5] | |
| One | 12.1 [10.5, 13.9] | |
| Two or more | 6.5 [3.8, 10.9] | |
| **Heard about other STIs** | | <0.001 |
| No | 7.8 [6.5, 9.3] | |
| Yes | 13.3 [12.1, 14.7] | |

*(Continued)*

**Table 2.** (Continued)

| Variables | Awareness of and positive attitudes toward PrEP | p-value |
|---|---|---|
| **Visited health facility last 12 months** | | <0.001 |
| No | 10.0 [8.9, 11.3] | |
| Yes | 14.0 [12.6, 15.6] | |
| **Wealth index** | | <0.001 |
| Poorest | 10.7 [9.0, 12.8] | |
| Poorer | 11.2 [9.7, 12.9] | |
| Middle | 11.4 [9.6, 13.4] | |
| Richer | 13.1 [11.2, 15.2] | |
| Richest | 13.4 [11.6, 15.5] | |
| **Place of residence** | | <0.001 |
| Urban | 12.9 [11.3, 14.7] | |
| Rural | 11.0 [9.9, 12.2] | |
| **Region** | | <0.001 |
| Western | 7.4 [5.3, 10.4] | |
| Central | 15.0 [12.1, 18.4] | |
| Greater Accra | 7.3 [5.1, 10.2] | |
| Volta | 18.8 [14.7, 23.7] | |
| Eastern | 10.8 [8.2, 14.1] | |
| Ashanti | 10.6 [7.9, 14.1] | |
| Western North | 14.0 [10.2, 18.8] | |
| Ahafo | 21.1 [15.7, 27.7] | |
| Bono | 15.9 [11.3, 21.9] | |
| Bono East | 12.0 [9.4, 15.0] | |
| Oti | 16.7 [13.2, 21.0] | |
| Northern | 18.8 [11.9, 28.4] | |
| Savannah | 7.2 [5.3, 9.9] | |
| North East | 9.7 [5.3, 17.2] | |
| Upper East | 15.1 [11.6, 19.4] | |
| Upper West | 11.6 [9.1, 14.7] | |

*P-values generated from testparm analysis

CI: 1.01–1.43). Having two or more partners was associated with lower odds (aOR = 0.51, 95% CI: 0.28–0.96) of having awareness of and positive attitudes toward PrEP than those with no partners. Women who had heard about other STIs had higher odds of awareness of and positive attitudes toward PrEP (aOR = 1.79, 95% CI: 1.40–2.28). Regional variations were also observed, with women from Central (aOR = 3.28, 95% CI: 1.76–6.12), Volta (aOR = 3.28, 95% CI: 1.63–6.62), Western North (aOR = 2.59, 95% CI: 1.26–5.29), Ahafo (aOR = 5.02, 95% CI: 2.54–9.91), Bono (aOR = 2.87, 95% CI: 1.37–6.02), Bono East (aOR = 2.87, 95% CI: 1.56–5.26), Oti (aOR = 5.64, 95% CI: 3.06–10.39), Northern (aOR = 2.47, 95% CI: 1.10–5.51), Upper East (aOR = 4.17, 95% CI: 2.05–8.50), and Upper West (aOR = 3.62, 95% CI: 1.85–7.08) regions having significantly higher odds of awareness of and positive attitudes toward PrEP than those in the Western region.

## Discussion

This study examined awareness of and positive attitudes toward PrEP for HIV prevention among women of reproductive age in Ghana. Only 12.1% of women were aware of and held positive attitudes toward PrEP, indicating low overall

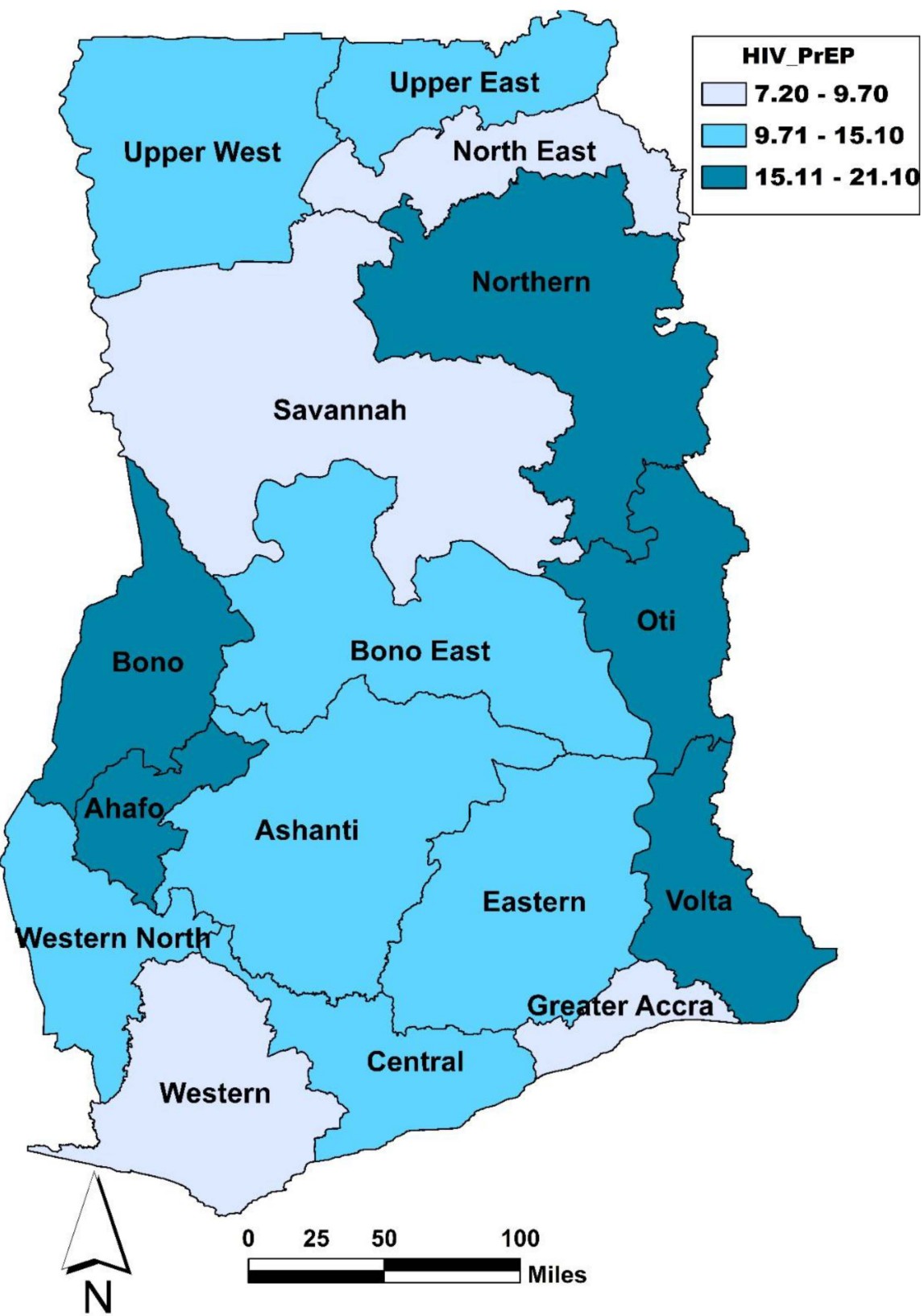

**Fig 1. Prevalence of awareness of and positive attitudes towards PrEP by region, Ghana, 2022. Source:** Author-generated, 2025.

**Table 3.** Factors associated with awareness of and positive attitudes toward PrEP among women in Ghana.

| Variables | Model I<br>Empty model | Model II<br>aOR [95% CI] | Model III<br>aOR [95% CI] | Model IV<br>aOR [95% CI] |
|---|---|---|---|---|
| **Fixed effect results** | | | | |
| **Woman's age (years)** | | | | |
| 15-19 | | 1.00 | | 1.00 |
| 20-24 | | 1.27 [0.94, 1.71] | | 1.28 [0.95, 1.73] |
| 25-29 | | 1.74** [1.27, 2.39] | | 1.75***[1.28, 2.39] |
| 30-34 | | 1.59*[1.14, 2.25] | | 1.61* [1.14, 2.27] |
| 35-39 | | 1.67* [1.21, 2.30] | | 1.69* [1.22, 2.32] |
| 40-44 | | 1.48* [1.03, 2.12] | | 1.48* [1.03, 2.12] |
| 45-49 | | 1.38 [0.93, 2.05] | | 1.39 [0.93, 2.07] |
| **Educational level** | | | | |
| No education | | 1.00 | | 1.00 |
| Primary | | 0.96 [0.74, 1.24] | | 0.96 [0.75, 1.24] |
| Secondary | | 1.35* [1.06, 1.71] | | 1.34* [1.06, 1.70] |
| Higher | | 3.34*** [2.43, 4.59] | | 3.27*** [2.35, 4.55] |
| **Marital status** | | | | |
| Never in union | | 1.00 | | 1.0 |
| Married | | 1.07 [0.78, 1.45] | | 1.06 [0.77, 1.44] |
| Cohabiting | | 0.99 [0.74, 1.34] | | 0.98 [0.73, 1.33] |
| Previously married | | 1.03 [0.75, 1.41] | | 1.03 [0.75, 1.41] |
| **Religion** | | | | |
| Christians | | 1.00 | | 1.00 |
| Muslims | | 0.99 [0.77, 1.26] | | 0.98 [0.77, 1.25] |
| Traditionalist | | 0.87 [0.48, 1.58] | | 0.87 [0.48, 1.57] |
| No religion | | 1.21 [0.56, 2.63] | | 1.23 [0.56, 2.69] |
| **Current working status** | | | | |
| Not working | | 1.00 | | 1.00 |
| Working | | 1.18 [0.98, 1.43] | | 1.18 [0.98, 1.43] |
| **Listen to radio** | | | | |
| No | | 1.00 | | 1.00 |
| Yes | | 1.20* [1.01, 1.43] | | 1.20* [1.01, 1.43] |
| **Watch television** | | | | |
| No | | 1.00 | | 1.00 |
| Yes | | 1.03 [0.86, 1.24] | | 1.02 [0.84, 1.24] |
| **Use internet** | | | | |
| No | | 1.00 | | 1.00 |
| Yes | | 0.95 [0.79, 1.15] | | 0.94 [0.78, 1.14] |
| **Number of partners excluding spouse** | | | | |
| Zero | | 1.00 | | 1.00 |
| One | | 0.99 [0.79, 1.26] | | 0.99 [0.79, 1.26] |
| Two or more | | 0.51 [0.28, 0.96] | | 0.51** [0.28, 0.96] |
| **Heard about other STIs** | | | | |
| No | | 1.00 | | 1.00 |
| Yes | | 1.79*** [1.40, 2.28] | | 1.79*** [1.40, 2.28] |
| **Visited health facility last 12 months** | | | | |
| No | | 1.00 | | 1.00 |

*(Continued)*

**Table 3.** (Continued)

| Variables | Model I Empty model | Model II aOR [95% CI] | Model III aOR [95% CI] | Model IV aOR [95% CI] |
|---|---|---|---|---|
| Yes | | 1.09 [0.93, 1.29] | | 1.09 [0.92, 1.28] |
| **Wealth index** | | | | |
| Poorest | | | 1.00 | 1.00 |
| Poorer | | | 1.19 [0.92, 1.53] | 1.10 [0.85, 1.44] |
| Middle | | | 1.27 [0.94, 1.69] | 1.03 [0.74, 1.44] |
| Richer | | | 1.76*** [1.31, 2.38] | 1.19 [0.84, 1.69] |
| Richest | | | 2.35*** [1.67, 3.29] | 1.16 [0.78, 1.74] |
| **Place of residence** | | | | |
| Urban | | | 1.00 | 1.00 |
| Rural | | | 0.79 [0.58, 1.07] | 0.78 [0.57, 1.06] |
| **Region** | | | | |
| Western | | | 1.00 | 1.00 |
| Central | | | 3.23*** [1.74, 6.01] | 3.28*** [1.76, 6.12] |
| Greater Accra | | | 0.53 [0.24, 1.14] | 0.53 [0.24, 1.18] |
| Volta | | | 3.39** [1.68, 6.84] | 3.28** [1.63, 6.62] |
| Eastern | | | 1.75* [0.92, 3.32] | 1.72 [0.89, 3.29] |
| Ashanti | | | 1.44 [0.76, 2.75] | 1.41 [0.74, 2.68] |
| Western North | | | 2.84* [1.39, 5.78] | 2.59* [1.26, 5.29] |
| Ahafo | | | 5.33*** [2.72, 10.43] | 5.02*** [2.54, 9.91] |
| Bono | | | 2.93* [1.43, 6.04] | 2.87* [1.37, 6.02] |
| Bono East | | | 2.86** [1.56, 5.24] | 2.87** [1.56, 5.26] |
| Oti | | | 5.57***[3.05, 10.17] | 5.64*** [3.06, 10.39] |
| Northern | | | 2.15* [1.002, 4.63] | 2.47* [1.10, 5.51] |
| Savannah | | | 1.31 [0.58, 2.95] | 1.55 [0.68, 3.56] |
| North East | | | 1.13 [0.47, 2.72] | 1.18 [0.48, 2.89] |
| Upper East | | | 4.55*** [2.28, 9.06] | 4.17*** [2.05, 8.50] |
| Upper West | | | 3.24*** [1.70, 6.19] | 3.62*** [1.85, 7.08] |
| **Random effect model** | | | | |
| PSU variance (95% CI) | 2.51 [1.97,3.19] | 2.63 [2.07, 3.33] | 2.14 [1.70, 2.69] | 2.19 [1.75, 2.74] |
| ICC | 0.433 | 0.444 | 0.394 | 0.399 |
| N | 13,980 | 13,980 | 13,980 | 13,980 |
| Number of PSUs | 618 | 618 | 618 | 618 |

aOR= adjusted odds ratios; CI = Confidence Interval; * p< 0.05, ** p< 0.01, *** p<0.001; 1.00 = Reference category; PSU=Primary Sampling Unit; ICC = Intra-Class Correlation Coefficient.

awareness and positive attitudes. This is consistent with earlier research among sexually active adults in Ghana [31]. However, the percentage reported in this study is higher than in Tanzania (5.61%), Burkina Faso (6.43%), and Côte d'Ivoire (9.56%), but lower than in Kenya (34.29%) [21,32]. These differences may be due to country-specific variations in PrEP awareness campaigns and their implementation. Kenya, a leader in the national rollout of PrEP since 2017, has focused its efforts on high-risk groups such as MSM, sex workers, and adolescent girls [33].

In Ghana, low awareness of and negative attitudes toward PrEP are concerning, particularly with increasing HIV infections [34]. Although PrEP acceptability has been reported as relatively high among MSM and healthcare providers in Ghana, actual use remains limited in both the general and key populations. Barriers include stigma, cost, and

misinformation [20,35,36]. Our study identified several factors associated with awareness of and positive attitudes toward PrEP, including women's age, educational level, radio exposure, number of sexual partners, knowledge of other STIs, and region of residence.

Older women had higher odds of being aware of and holding positive attitudes toward PrEP than adolescents, consistent with findings by Terefe et al. [21] across five sub-Saharan countries. Older women are more likely to attend antenatal clinics where HIV education is provided, increasing their exposure to prevention methods like PrEP [37]. Conversely, adolescent girls face numerous social and structural barriers, such as stigma surrounding adolescent sexuality and disapproving attitudes from healthcare providers. In Ghana, conversations about sex are often taboo, and young girls seeking information or services for HIV prevention may be perceived as promiscuous [38–40]. Given the high vulnerability of adolescent girls and young women to HIV [41], interventions should prioritise them through targeted education and the integration of comprehensive sexuality education into programmes like the School Health Education Programme (SHEP), to empower young girls with the knowledge and confidence needed to protect themselves.

Women with secondary or higher education had higher odds of being aware of and having positive attitudes toward PrEP. This finding aligns with previous studies [21,42]. Bailey et al. reported no significant association between education and PrEP knowledge among nurses in South Africa [43]. Higher education improves women's access to healthcare and health information, including comprehensive knowledge of HIV/AIDS [21,36,44,45], potentially influenced by internet access, community groups, and health promotion initiatives. However, these mixed results may stem from differences in study populations and contexts. In the general population, higher educational levels often lead to better access to health information and more proactive health-seeking behaviour [21,44,46], whereas among healthcare professionals such as nurses, standardised clinical training and workplace exposure may be more influential. These contrasting findings highlight the need for context-specific strategies to enhance PrEP awareness and foster positive attitudes toward its use.

Radio exposure was also positively associated with awareness of and positive attitudes toward PrEP, supporting findings by Terefe et al. [21]. Asamoah et al. [47] further reported that exposure to mass media enhances HIV knowledge and lessens stigmatising attitudes in Ghana. Radio remains one of the most accessible media in Ghana, particularly in rural areas, with content often delivered in local languages [48,49]. It plays a crucial role in spreading health information, including HIV prevention education [49]. However, challenges persist, such as language barriers and limited access to modern communication tools [48]. By using culturally sensitive and empowering messaging, radio can demystify PrEP and dispel misconceptions. Nonetheless, caution is necessary to ensure these campaigns do not unintentionally reinforce harmful gender norms or stigmatise already marginalised women. This underscores the importance of collaboration between public health actors and media professionals [50].

Women who had heard about other STIs had higher odds of awareness of and positive attitudes toward PrEP. This agrees with existing literature showing that prior exposure to STI-related information increases PrEP awareness and acceptance [21]. Such information often comes from health campaigns or media, indicating that wider sexual health messaging can positively influence PrEP-related outcomes.

Women with two or more sexual partners had lower odds of PrEP awareness and positive attitudes toward it, despite being at higher risk for HIV [51]. This aligns with findings from Malawi by Kabaghe et al. [22]. While it might be expected that high-risk individuals would be more informed, this counterintuitive result may reflect the stigma, fear, and social judgment surrounding PrEP use. Studies show that women worry about being perceived as promiscuous or HIV-positive if they take PrEP [52], and they may anticipate disapproval from partners, family, or healthcare providers [53]. Furthermore, healthcare systems may not effectively reach women with complex sexual networks due to insufficiently tailored outreach. These findings emphasise the need for inclusive, stigma-free sexual health education that provides high-risk women with accurate and nonjudgmental information about PrEP.

We also found notable regional disparities in awareness of and positive attitudes toward PrEP. Women in the Ahafo, Oti, and Upper East regions had significantly higher odds of being aware of and holding positive attitudes toward PrEP than those in the Western region. This reflects earlier findings by Apreku et al. [27] on regional differences in PrEP awareness among Ghanaian MSM. Bono (formerly part of Brong Ahafo), Oti (formerly part of Volta), and Upper East regions have also been recognised for having fewer barriers to healthcare access [54]. Variations in healthcare infrastructure, cultural norms, geographical access, and socioeconomic factors may account for these disparities. Furthermore, the presence of active Non-governmental Organizations (NGOs) and local campaigns could also play a role. For example, MIHOSO International Foundation and partners have conducted HIV prevention campaigns in Ahafo [55]. These findings imply that strengthening health systems, investing in region-specific community outreach, and collaborating with local NGOs could enhance PrEP outcomes across the country.

## Strengths and limitations of the study

The use of nationally representative survey data is a key strength of this study. It ensures representativeness at the national level and improves the generalisability of our findings to all women of reproductive age in Ghana. However, the study has some limitations that are worth noting. First, due to the cross-sectional design nature of the GDHS, a cause-and-effect relationship between the outcome and exposure variables cannot be established. Second, self-reported awareness of and positive attitudes toward PrEP may be vulnerable to social desirability bias or recall bias, which could affect response accuracy. By combining awareness and approval into a single measure, this study could not differentiate the determinants of awareness from those influencing approval, potentially impacting the strength and direction of the associations observed with socio-demographic factors. Furthermore, other factors that might influence awareness of and positive attitudes toward PrEP, such as sociocultural influences, issues with health professionals, and support programmes, were not included because data on these variables were unavailable from the GDHS.

## Conclusions

Approximately 1 in 10 women aged 15–49 years in Ghana were aware of PrEP and had positive attitudes toward it. Factors such as women's age, educational level, number of sexual partners, listening to the radio, prior awareness of other STIs, and geographical region are significantly associated with women's awareness of and positive attitudes toward PrEP. This reveals a gap in awareness of and positive attitudes toward PrEP among women in Ghana, highlighting the need for targeted educational campaigns and equitable access to HIV prevention services. Policymakers and healthcare providers should focus on addressing regional disparities and barriers to PrEP awareness and uptake to enhance HIV prevention efforts across the country. In the planning and implementation of HIV and PrEP programmes, it is essential to consider both age and educational level to ensure that interventions are appropriately tailored, accessible, and effective for the targeted populations. Mass media campaigns should be utilised to raise awareness of PrEP. Furthermore, increasing awareness among the general population in Ghana, beyond key populations, is crucial to reducing new infections and meeting the Sustainable Development Goal 3, target 3.3. Future research should examine awareness and approval of PrEP as separate outcomes. Awareness is mainly influenced by exposure to information and access to healthcare, while approval is shaped by risk perception, attitudes, and cultural norms. Distinguishing these factors in future studies will enable more targeted and effective interventions to boost PrEP uptake.

## Author contributions

**Conceptualization:** Florence Gyembuzie Wongnaah, Richard Gyan Aboagye.

**Data curation:** Florence Gyembuzie Wongnaah, Richard Gyan Aboagye.

**Formal analysis:** Florence Gyembuzie Wongnaah, Richard Gyan Aboagye.

**Supervision:** Richard Gyan Aboagye.

**Writing – original draft:** Florence Gyembuzie Wongnaah, Gilbert Eshun, Mainprice Akuoko Essuman, Collins Adu.

**Writing – review & editing:** Florence Gyembuzie Wongnaah, Gilbert Eshun, Mainprice Akuoko Essuman, Collins Adu, Richard Gyan Aboagye.

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
