## [Decision Letter · Decision Letter 0]

11 Dec 2025

PONE-D-25-54678Knowledge and attitudes toward HIV Pre-Exposure prophylaxis among reproductive-aged women in Ghana: A Multilevel Analysis of the 2022 Demographic and Health SurveyPLOS One

Dear Dr. Wongnaah,

Thank you for submitting your manuscript to PLOS ONE. After careful consideration, we feel that it has merit but does not fully meet PLOS ONE’s publication criteria as it currently stands. Therefore, we invite you to submit a revised version of the manuscript that addresses the points raised during the review process.

If applicable, we recommend that you deposit your laboratory protocols in protocols.io to enhance the reproducibility of your results. Protocols.io assigns your protocol its own identifier (DOI) so that it can be cited independently in the future. For instructions see: https://journals.plos.org/plosone/s/submission-guidelines#loc-laboratory-protocols. Additionally, PLOS ONE offers an option for publishing peer-reviewed Lab Protocol articles, which describe protocols hosted on protocols.io. Read more information on sharing protocols at . Additionally, PLOS ONE offers an option for publishing peer-reviewed Lab Protocol articles, which describe protocols hosted on protocols.io. Read more information on sharing protocols at https://plos.org/protocols?utm_medium=editorial-email&utm_source=authorletters&utm_campaign=protocols..

We look forward to receiving your revised manuscript.

Kind regards,

Anthony Mwinilanaa Tampah-Naah

Academic Editor

PLOS One

Journal Requirements:

2. Please note that your Data Availability Statement is currently missing the DOI/accession number of each dataset OR a direct link to access each database. If your manuscript is accepted for publication, you will be asked to provide these details on a very short timeline. We therefore suggest that you provide this information now, though we will not hold up the peer review process if you are unable.

3. We note that Figure 1 in your submission contains a map image which may be copyrighted. All PLOS content is published under the Creative Commons Attribution License (CC BY 4.0), which means that the manuscript, images, and Supporting Information files will be freely available online, and any third party is permitted to access, download, copy, distribute, and use these materials in any way, even commercially, with proper attribution. For these reasons, we cannot publish previously copyrighted maps or satellite images created using proprietary data, such as Google software (Google Maps, Street View, and Earth). For more information, see our copyright guidelines: http://journals.plos.org/plosone/s/licenses-and-copyright.

Reviewers' comments:

Reviewer's Responses to Questions

**Comments to the Author**

1. Is the manuscript technically sound, and do the data support the conclusions?

Reviewer #1: Yes

Reviewer #2: Yes

2. Has the statistical analysis been performed appropriately and rigorously? 

Reviewer #1: Yes

Reviewer #2: Yes

3. Have the authors made all data underlying the findings in their manuscript fully available?

Reviewer #1: Yes

Reviewer #2: Yes

4. Is the manuscript presented in an intelligible fashion and written in standard English?

Reviewer #1: Yes

Reviewer #2: Yes

5. Review Comments to the Author

Reviewer #1: Comment

Well written manuscript on Knowledge and attitudes toward HIV Pre-Exposure prophylaxis among reproductive aged women in Ghana: A Multilevel Analysis of the 2022 Demographic and Health Survey.

A few comments:

Abstract

1. It is important to describe in the methods how the questions explored knowledge and attitude. Who is knowledgeable? What do you mean by attitude (good/bad?). One/two sentences on how these variables were measured should suffice

2. Report higher or lower odds of outcome. Odds ratio is not a risk ratio. Do not use word like more likely/less likely, likelihood when using Odds ratio. These terminologies should be used when interpreting Risk ratio, Hazard Ratio, and Incidence Rate Ratio. This comment should be cross checked throughout the manuscript.

3. Up until reading the conclusion, results start making sense when you say positive attitude .Otherwise it was not flowing well when you say have knowledge and attitude. It should in fact be good knowledge and positive attitude if you like. Please make correction throughout the abstract.

Main text

1. Would you please share the reason(s) of using multilevel modelling in the analysis section?

2. Is wealth index a community variable? It does not change results, but I am just concerned treating it as community variable. I would understand place of residence and regions where participants are nested.

3. Please edit title of figure 1: Is it PrEP awareness prevalence? PrEP awareness and positive attitude prevalence. It should be clear

4. Line 81, define MSM, it seems that it is the first time it appears in the main text

5. Line 255, 258 etc. Once defined and abbreviated, use abbreviations throughout. I see PrEP being defined again and MSM was used earlier but now defined.

6. Line 258-259: Your study was among reproductive aged women from general population. It would be better to compare results with studies of similar population instead of using MSM, TGW as comparison groups.

Reviewer #2: Summary

The authors analyzed date from 13, 980 women ages 15-49 who participated in the GDHS and provided information about their prior knowledge of PrEP and approval, disapproval or indifference to its use. Findings were reported dichotomously with Participants who reported having heard about PrEP and approved of its use being classified as knowledgeable and having favorable attitude towards PrEP then coded as “Yes=1”. Participants who had not heard about PrEP or did not express a positive attitude towards its use were coded as “No=0”. The proportion of women who knew about PrEP and approved of its use was low at 12.3% . The authors also identified factors associated with awareness and positive attitude towards PrEP.

General Comments

The manuscript is well written and addresses an important public health topic; however, it could benefit from revisions clarifying that the survey questions assessed participants’ awareness of PrEP and their approval of its use. Throughout the paper, awareness and knowledge appear to be used interchangeably; this may confuse some readers. Differentiating awareness (having heard about PrEP) and knowledge (understanding its purpose, effectiveness, benefits, and appropriate use) could improve clarity.

Specific comments

Abstract

Background (Line 27-29)

Consider splitting the single sentence under background into two or separate the two parts using a semi colon for easy reading.

Methods

If the word limit allows, include a sentence which explains how knowledge and attitude were measured for benefit of readers who choose not to read the whole manuscript.

Results (Line 35-36)

Consider editing the first sentence under results to indicate whether the groups of interest were women who had never heard about PrEP and those already know about PrEP. Similarly for attitude, it should be clarified that the groups of interest are those who approved of PrEP use and those who did not. those who expressed no attitude (i.e., indifferent) vs those who had some attitude (whether negative or positive).

The authors may consider reframing the opening sentence in the results section of the abstract to read as follows; Overall, 12.13% of women reported prior knowledge of PrEP and approved of its use….

Conclusion (line 46-47)

Consider editing the conclusion in the abstract section to align with the suggestions already given for results above and to conventional scientific reporting.

For Example, approximately one in ten women in Ghana demonstrated adequate knowledge and positive attitudes towards PrEP…..

Background (line 99-101)

It stated in part that …… knowledge and attitudes toward PrEP remain critical barriers to

its adoption and acceptance…, include

Consider editing to read as follows;

Despite its proven efficacy, lack of knowledge of and negative attitudes towards PREP remain significant barriers to its adoption…

Methods (Line 145-148)

States that “Participants responded by indicating whether they had never heard of PrEP, heard about it, heard and approved of taking PrEP every day, heard but did not approve of taking PrEP the day before, or heard of PrEP but not sure about approving it.” Review this sentence to ascertain that there are no misplaced words. Taking PrEP the day before appears misplaced in this context.

Results

Line 213: States that the overall proportion of women with good knowledge and positive

attitudes toward PrEP were 12.13%. Consider removing the word good from this sentence unless the quality of knowledge was assessed.

6. PLOS authors have the option to publish the peer review history of their article (what does this mean?). If published, this will include your full peer review and any attached files.). If published, this will include your full peer review and any attached files.

.

Reviewer #1: No

Reviewer #2: No

---

## [Author Response · Author response to Decision Letter 1]

21 Jan 2026

RESPONSE TO REVIEWERS’ COMMENTS

Reviewer #1: Comment

Well written manuscript on Knowledge and attitudes toward HIV Pre-Exposure prophylaxis among reproductive aged women in Ghana: A Multilevel Analysis of the 2022 Demographic and Health Survey.

A few comments:

Abstract

1. It is important to describe in the methods how the questions explored knowledge and attitude. Who is knowledgeable? What do you mean by attitude (good/bad?). One/two sentences on how these variables were measured should suffice

Response: The outcome variable was knowledge and positive attitude toward PrEP, assessed based on whether respondents had heard of PrEP and approved of people taking PrEP to prevent acquiring HIV.

2. Report higher or lower odds of outcome. Odds ratio is not a risk ratio. Do not use word like more likely/less likely, likelihood when using Odds ratio. These terminologies should be used when interpreting Risk ratio, Hazard Ratio, and Incidence Rate Ratio. This comment should be cross checked throughout the manuscript.

Response: We have revised the manuscript to ensure that all interpretations of the regression results are consistent with odds ratios. Specifically, we replaced terms such as “more likely,” “less likely,” and “likelihood” with “higher odds” or “lower odds” where odds ratios are reported. This correction has been applied across the abstract, results, discussion, and conclusion sections.

3. Up until reading the conclusion, results start making sense when you say positive attitude. Otherwise it was not flowing well when you say have knowledge and attitude. It should in fact be good knowledge and positive attitude if you like. Please make correction throughout the abstract.

Response: To improve clarity and flow, we have revised the manuscript to consistently use the term “knowledge and positive attitude toward PrEP” across the abstract and main text.

Main text

1. Would you please share the reason(s) of using multilevel modelling in the analysis section?

Response: Multilevel logistic regression was used to account for the hierarchical structure of the DHS data, in which women are nested within households and communities. This approach allows for the estimation of both individual- and community-level effects while adjusting for intra-cluster correlation.

2. Is wealth index a community variable? It does not change results, but I am just concerned treating it as community variable. I would understand place of residence and regions where participants are nested.

Response: Wealth index is a household variable. It is created as a composite variable from several items available in the households where a woman resides. Wealth index, together with place and region of residence, was grouped as contextual-level variables used in the current study.

3. Please edit title of figure 1: Is it PrEP awareness prevalence? PrEP awareness and positive attitude prevalence. It should be clear

Response: We have revised the title of Figure 1 to reflect both components of the outcome variable clearly. The updated title now reads: “HIV PrEP awareness and positive attitude prevalence by region, Ghana, 2022.”

4. Line 81, define MSM, it seems that it is the first time it appears in the main text

Response: We have revised the manuscript to define men who have sex with men (MSM) at its first occurrence in the main text.

5. Line 255, 258 etc. Once defined and abbreviated, use abbreviations throughout. I see PrEP being defined again and MSM was used earlier but now defined.

Response: Thank you for this observation. We have carefully revised the manuscript to ensure that all terms are defined only at their first occurrence and that abbreviations (e.g., PrEP and MSM) are used consistently throughout the manuscript without redefinition.

6. Line 258-259: Your study was among reproductive aged women from general population. It would be better to compare results with studies of similar population instead of using MSM, TGW as comparison groups.

Response: We have revised the manuscript to compare our findings with studies conducted among sexually active adults from the general population. References to MSM and transgender women (TGW) have been removed.

Reviewer #2: Summary

The authors analyzed date from 13, 980 women ages 15-49 who participated in the GDHS and provided information about their prior knowledge of PrEP and approval, disapproval or indifference to its use. Findings were reported dichotomously with Participants who reported having heard about PrEP and approved of its use being classified as knowledgeable and having favorable attitude towards PrEP then coded as “Yes=1”. Participants who had not heard about PrEP or did not express a positive attitude towards its use were coded as “No=0”. The proportion of women who knew about PrEP and approved of its use was low at 12.3% . The authors also identified factors associated with awareness and positive attitude towards PrEP.

Response: Thank you.

General Comments

The manuscript is well written and addresses an important public health topic; however, it could benefit from revisions clarifying that the survey questions assessed participants’ awareness of PrEP and their approval of its use. Throughout the paper, awareness and knowledge appear to be used interchangeably; this may confuse some readers. Differentiating awareness (having heard about PrEP) and knowledge (understanding its purpose, effectiveness, benefits, and appropriate use) could improve clarity.

Response: The outcome variable is now consistently described as awareness of PrEP and positive attitudes toward its use, in line with the DHS Guide to Statistics. We have clarified that awareness refers to having heard of PrEP, while approval reflects participants’ positive attitude toward its use. The terminology has been revised throughout the manuscript, including the abstract, results, and discussion.

Specific comments

Abstract

Background (Line 27-29)

Consider splitting the single sentence under background into two or separate the two parts using a semi colon for easy reading.

Response: Thank you for this suggestion. We have revised the sentence by splitting it into two sentences, making the background clearer and easier to follow.

Methods

If the word limit allows, include a sentence which explains how knowledge and attitude were measured for benefit of readers who choose not to read the whole manuscript.

Response: We have added a sentence to clearly explain how the outcome was measured: The outcome variable was knowledge and positive attitude toward PrEP, assessed based on whether respondents had heard of PrEP and approved of people taking PrEP to prevent acquiring HIV.

Results (Line 35-36)

Consider editing the first sentence under results to indicate whether the groups of interest were women who had never heard about PrEP and those already know about PrEP. Similarly for attitude, it should be clarified that the groups of interest are those who approved of PrEP use and those who did not. those who expressed no attitude (i.e., indifferent) vs those who had some attitude (whether negative or positive).

Response: In line with our dichotomous outcome definition (knowledge = having heard of PrEP; positive attitude = approving of people taking PrEP), we have revised the Results sentence in the abstract to clearly report the prevalence of women with knowledge of PrEP and positive attitudes toward its use. Specifically, the sentence now reads: Overall, 12.13% (95% CI: 11.1–13.3) of women had knowledge of PrEP and positive attitudes toward its use.”

The authors may consider reframing the opening sentence in the results section of the abstract to read as follows; Overall, 12.13% of women reported prior knowledge of PrEP and approved of its use….

Response: We have retained the phrasing in the abstract as “Overall, 12.13% of women had knowledge of PrEP and approved of its use”, as it accurately reflects our outcome definition and aligns with DHS terminology.

Conclusion (line 46-47)

Consider editing the conclusion in the abstract section to align with the suggestions already given for results above and to conventional scientific reporting. For Example, approximately one in ten women in Ghana demonstrated adequate knowledge and positive attitudes towards PrEP…..

Response: We have revised the abstract conclusion to align with the updated Results phrasing. The revised sentence now reads: “About one in ten women in Ghana had knowledge of PrEP and positive attitudes toward its use.

Background (line 99-101)

It stated in part that …… knowledge and attitudes toward PrEP remain critical barriers to

its adoption and acceptance…, include

Consider editing to read as follows;

Despite its proven efficacy, lack of knowledge of and negative attitudes towards PREP remain significant barriers to its adoption…

Response: We have revised the sentence in the background to improve clarity. The sentence now reads: “Despite its proven efficacy, lack of knowledge of and negative attitudes toward PrEP remain significant barriers to its adoption and acceptance.”

Methods (Line 145-148)

States that “Participants responded by indicating whether they had never heard of PrEP, heard about it, heard and approved of taking PrEP every day, heard but did not approve of taking PrEP the day before, or heard of PrEP but not sure about approving it.” Review this sentence to ascertain that there are no misplaced words. Taking PrEP the day before appears misplaced in this context.

Response: We have revised the sentence to remove misplaced phrases and better reflect the DHS survey response options. The revised sentence now reads: Participants responded by indicating whether they had never heard of PrEP, had heard of PrEP, had heard of PrEP and approved of people taking it, had heard of PrEP but did not approve of its use, or had heard of PrEP but were unsure about approving it.

Results

Line 213: States that the overall proportion of women with good knowledge and positive

attitudes toward PrEP were 12.13%. Consider removing the word good from this sentence unless the quality of knowledge was assessed.

Response: The word “good” has been removed to accurately reflect that our outcome measured whether women had knowledge of PrEP, rather than the quality of knowledge. The revised sentence now reads: The overall proportion of women with knowledge of PrEP and positive attitudes toward its use was 12.13%.

3 We note that Figure 1 in your submission contains a map image which may be copyrighted. All PLOS content is published under the Creative Commons Attribution License (CC BY 4.0), which means that the manuscript, images, and Supporting Information files will be freely available online, and any third party is permitted to access, download, copy, distribute, and use these materials in any way, even commercially, with proper attribution. For these reasons, we cannot publish previously copyrighted maps or satellite im-ages created using proprietary data, such as Google software (Google Maps, Street View, and Earth). For more information, see our copyright guidelines: http://journals.plos.org/plosone/s/licenses-and-copyright.

1. You may seek permission from the original copyright holder of Figure 1 to publish the content specifi-cally under the CC BY 4.0 license.

“I request permission for the open-access journal PLOS ONE to publish XXX under the Creative Com-mons Attribution License (CCAL) CC BY 4.0 (http://creativecommons.org/licenses/by/4.0/). Please be aware that this license allows unrestricted use and distribution, even commercially, by third parties. Please reply and provide explicit written permission to publish XXX under a CC BY license and complete the at-tached form.”

Please upload the completed Content Permission Form or other proof of granted permissions as an "Oth-er" file with your submission.

In the figure caption of the copyrighted figure, please include the following text: “Reprinted from [ref] un-der a CC BY license, with permission from [name of publisher], original copyright [original copyright year].”

2. If you are unable to obtain permission from the original copyright holder to publish these figures under the CC BY 4.0 license or if the copyright holder’s requirements are incompatible with the CC BY 4.0 li-cense, please either i) remove the figure or ii) supply a replacement figure that complies with the CC BY 4.0 license. Please check copyright information on all replacement figures and update the figure caption with source information. If applicable, please specify in the figure caption text when a figure is similar but not identical to the original image and is therefore for illustrative purposes only.

Response: Thank you for raising concerns regarding the copyright status of Figure 1 in our manuscript. We want to clarify that Figure 1 was entirely generated by the authors using Geographic Information System (GIS) software and does not contain any copyrighted or proprietary map images. The figure was created using publicly available, open-access shape files and author-generated analytical outputs. No data, imagery, or base maps from proprietary sources such as Google Maps, Google Earth, or other restricted platforms were used. Therefore, no additional permissions or licenses are required for its publication.

---

## [Decision Letter · Decision Letter 1]

22 Feb 2026

PONE-D-25-54678R1Awareness of HIV Pre-Exposure Prophylaxis and Positive Attitudes Among Reproductive-Aged Women in Ghana: A Multilevel Analysis of the 2022 Demographic and Health SurveyPLOS One

Dear Dr. Wongnaah,

Thank you for submitting your manuscript to PLOS ONE. After careful consideration, we feel that it has merit but does not fully meet PLOS ONE’s publication criteria as it currently stands. Therefore, we invite you to submit a revised version of the manuscript that addresses the points raised during the review process.

If applicable, we recommend that you deposit your laboratory protocols in protocols.io to enhance the reproducibility of your results. Protocols.io assigns your protocol its own identifier (DOI) so that it can be cited independently in the future. For instructions see: https://journals.plos.org/plosone/s/submission-guidelines#loc-laboratory-protocols. Additionally, PLOS ONE offers an option for publishing peer-reviewed Lab Protocol articles, which describe protocols hosted on protocols.io. Read more information on sharing protocols at . Additionally, PLOS ONE offers an option for publishing peer-reviewed Lab Protocol articles, which describe protocols hosted on protocols.io. Read more information on sharing protocols at https://plos.org/protocols?utm_medium=editorial-email&utm_source=authorletters&utm_campaign=protocols..

We look forward to receiving your revised manuscript.

Kind regards,

Anthony Mwinilanaa Tampah-Naah

Academic Editor

PLOS One

Journal Requirements:

Reviewers' comments:

Reviewer's Responses to Questions

**Comments to the Author**

1. If the authors have adequately addressed your comments raised in a previous round of review and you feel that this manuscript is now acceptable for publication, you may indicate that here to bypass the “Comments to the Author” section, enter your conflict of interest statement in the “Confidential to Editor” section, and submit your "Accept" recommendation.

Reviewer #1: All comments have been addressed

Reviewer #2: All comments have been addressed

Reviewer #3: All comments have been addressed

2. Is the manuscript technically sound, and do the data support the conclusions?

Reviewer #1: Yes

Reviewer #2: Yes

Reviewer #3: Yes

3. Has the statistical analysis been performed appropriately and rigorously? 

Reviewer #1: Yes

Reviewer #2: Yes

Reviewer #3: Yes

4. Have the authors made all data underlying the findings in their manuscript fully available?

Reviewer #1: Yes

Reviewer #2: Yes

Reviewer #3: Yes

5. Is the manuscript presented in an intelligible fashion and written in standard English?

Reviewer #1: Yes

Reviewer #2: Yes

Reviewer #3: Yes

6. Review Comments to the Author

Reviewer #1: Authors have addressed my previous comments. The reason I am recommending minor revision, I would like them to read it one more time. When you track change and accept tracked changes, there are a few sentences don't rhyme.

For example in the abstract line 43 (....Northen had region the lowest.."). This is just one of them, could be more sentences of this type.

Making authors read their work before resubmission will prevent these errors.

Job well done.

Reviewer #2: The authors have adequately addressed comments raised in the previous round of review. I consider the manuscript acceptable for publication. No additional comments.

Reviewer #3: This revised manuscript has been improved substantially in terms of wording consistency and clarity. But fewer questions and comments that authors need to further address and specify throughout the manuscript before it could be considered for publication. I will enclose it.

7. PLOS authors have the option to publish the peer review history of their article (what does this mean?). If published, this will include your full peer review and any attached files.). If published, this will include your full peer review and any attached files.

.

Reviewer #1: No

Reviewer #2: No

Reviewer #3: No

---

## [Author Response · Author response to Decision Letter 2]

18 Mar 2026

1

Review date February 09, 2026

Journal PLOS ONE

Manuscript ID PONE-D-25-54678R1

Awareness of HIV Pre-Exposure Prophylaxis and Positive Attitudes among Reproductive-Aged Women in Ghana: A Multilevel Analysis of the 2022 Demographic and Health Survey

Reviewer's report

This study examined the factors associated with PrEP awareness and positive attitudes toward its use among women of reproductive age in Ghana. The authors reported that, overall, 12.13% of women were aware of PrEP and had positive attitudes toward its use. The Ashanti region had the highest proportion, 20.25%, whereas the Northern region had the lowest. The main predictors of PrEP awareness and positive attitudes toward its use included being aged 25–29 years, 30–39 years, and 40–44 years; having heard about STIs; reporting higher education; listening to the radio; and residing in the Ahafo region, Oti region, and Upper East region. However, those with ≥ two sexual partners had lower awareness and positive attitudes toward PrEP.

This revised manuscript has been improved substantially in terms of wording consistency and clarity. But I still have fewer questions and comments that authors need to further address and specify throughout the manuscript before it could be considered for publication. Here are the questions, comments and specifications that could be additionally addressed or revised to improve the clarity and content of the manuscript.

Abstract

1 - Results, please recheck sentences and figures with Table 1 for consistency. Authors should be clearer such as “...The Ashanti region had the highest proportion of sample distribution, 20.24%, whereas the North‒East had the lowest, 1.68%” rather than “...The Ashanti region had the highest proportion, 20.25%, whereas the Northern had the lowest, 1.68%”.

Response:

Thank you for this observation. The proportions reported refer to PrEP awareness and positive attitudes, which correspond to Table 2 rather than Table 1. However, we agree that the values reported in the abstract were inconsistent with the table. The abstract has now been revised to ensure that all figures correspond exactly with those reported in Table 2.

Background

1 - Paragraph: Line 118-123: It should add few citations about HIV PrEP awareness and positive attitudes toward its use in different regions such as Asia, SSA among women rather than citing among the MSM and sex worker.

Response: Thank you for this helpful suggestion. We revised the background section to include studies on PrEP awareness and positive attitudes among women and general populations in sub-Saharan Africa, specifically citing evidence from a multi-country study among women and a population-based study from Malawi [21,22]. We retained the Ghana studies among MSM and FSWs to provide local context, while clarifying that little is known about PrEP awareness and positive attitudes among women in the general population in Ghana.

Methods

1 - Line 146-147 “The Ghana Health Service Ethics Review Committee approved the study” should be moved to Ethical consideration section.

Response: Thank you for this observation. We have moved the statement “The Ghana Health Service Ethics Review Committee approved the study” from the Study design and description section to the Ethical consideration section for better organisation and clarity.

1 - It is not clear to me about outcome variable, could the authors detail more such as “How many questions are asked about awareness and how many about attitudes?” Any cut-off point defined as awareness and having positive attitudes toward PrEP?

Response: Thank you for this important comment. We have revised the description of the outcome variable in the Methods section to provide more clarity. Specifically, PrEP awareness and attitudes were assessed using questions from the 2022 Ghana Demographic and Health Survey (GDHS) women’s questionnaire. Respondents were first asked whether they had ever heard of HIV pre-exposure prophylaxis (PrEP). Those who answered “yes” were asked a follow-up question regarding whether they approved of people taking PrEP to prevent HIV infection.

To construct the outcome variable, we created a composite binary measure. Respondents who reported that they had heard of PrEP and approved of people taking PrEP to prevent HIV infection were classified as having PrEP awareness and a positive attitude (coded = 1). All other responses, including those who had never heard of PrEP, did not approve of its use, or were unsure, were coded as 0. The Methods section has been revised accordingly to clarify the number of questions used and the cut-off criteria applied in the analysis.

2 -

3 - Explanatory variables: Could you specify what variables were included in an individual level and what variables were included in a cluster level or community level since you used the multilevel analysis?

Response:

Thank you for this helpful comment. We have revised the Methods section to clearly distinguish between the individual-level and contextual (household and community-level) variables included in the multilevel analysis. Individual-level variables included women’s age, educational attainment, marital status, religion, employment status, exposure to mass media (reading newspaper or magazine, listening to radio, watching television, and internet use), number of sexual partners excluding spouse, awareness of other sexually transmitted infections (STIs), and whether the respondent visited a health facility in the past 12 months. The contextual-level variables included the wealth index, place of residence (urban/rural), and region. These clarifications have now been added to the “Explanatory variables” subsection of the Methods section.

Results

1 - The region variable was divided into 17 geographical regions. Is it a nationally pre-defined region? If not, some regions should be combined together due to uneven disproportionate distributions, such as the Ashanti region, which has a large sample distribution (20.24%), compared to Savannah (1.85%), North-East (1.68%) and others.

Response:

Thank you for this valuable comment. We would like to clarify that the regional variable used in this study reflects the 16 officially recognised administrative regions of Ghana, as defined following the 2019 regional reorganisation and captured in the 2022 Ghana Demographic and Health Survey (GDHS). These regions are nationally predefined administrative units and form part of the DHS sampling framework. The unequal distribution of respondents across regions reflects differences in population size and sampling allocation across the country. However, the DHS employs a complex sampling design and sampling weights to ensure national representativeness. These weights were applied in our analysis to account for disproportionate sampling. Additionally, retaining the official regional categories allows for more meaningful geographic interpretation of the findings and supports policy-relevant recommendations. For these reasons, we retained the 16 administrative regions as separate categories rather than combining them.

References

1 - It would be better to shorten numbers of reference citations down to about 35 or less since it is too much references (56) for such an research article.

Response:

Thank you for this helpful suggestion. We carefully reviewed the reference list and ensured that all cited studies are directly relevant to the background, methodology, and discussion of the study. Given the need to provide adequate context on HIV prevention and PrEP research in sub-Saharan Africa and Ghana, we retained the references that are essential to support the arguments presented in the manuscript. However, we have checked the list to remove any redundant citations where possible.

I declare that I have no conflicting interest in reviewing this manuscript.

---

## [Decision Letter · Decision Letter 2]

29 Mar 2026

Awareness of HIV Pre-Exposure Prophylaxis and Positive Attitudes Among Reproductive-Aged Women in Ghana: A Multilevel Analysis of the 2022 Demographic and Health Survey

PONE-D-25-54678R2

Dear Dr. Wongnaah,

We’re pleased to inform you that your manuscript has been judged scientifically suitable for publication and will be formally accepted for publication once it meets all outstanding technical requirements.

An invoice will be generated when your article is formally accepted. Please note, if your institution has a publishing partnership with PLOS and your article meets the relevant criteria, all or part of your publication costs will be covered. Please make sure your user information is up-to-date by logging into Editorial Manager at Editorial Manager® and clicking the ‘Update My Information' link at the top of the page. For questions related to billing, please contact  and clicking the ‘Update My Information' link at the top of the page. For questions related to billing, please contact billing support..

Kind regards,

Anthony Mwinilanaa Tampah-Naah

Academic Editor

PLOS One

Additional Editor Comments (optional):

Reviewers' comments:

Reviewer's Responses to Questions

**Comments to the Author**

1. If the authors have adequately addressed your comments raised in a previous round of review and you feel that this manuscript is now acceptable for publication, you may indicate that here to bypass the “Comments to the Author” section, enter your conflict of interest statement in the “Confidential to Editor” section, and submit your "Accept" recommendation.

Reviewer #1: All comments have been addressed

2. Is the manuscript technically sound, and do the data support the conclusions?

Reviewer #1: Yes

3. Has the statistical analysis been performed appropriately and rigorously? 

Reviewer #1: Yes

4. Have the authors made all data underlying the findings in their manuscript fully available?

Reviewer #1: Yes

5. Is the manuscript presented in an intelligible fashion and written in standard English?

Reviewer #1: Yes

6. Review Comments to the Author

Reviewer #1: No additional comments. All prior comments have been addressed. All prior comments have been addressed.

7. PLOS authors have the option to publish the peer review history of their article (what does this mean?). If published, this will include your full peer review and any attached files.). If published, this will include your full peer review and any attached files.

.

Reviewer #1: No

---

## [Editor Report · Acceptance letter]

PONE-D-25-54678R2

PLOS One

Dear Dr. Wongnaah,

I'm pleased to inform you that your manuscript has been deemed suitable for publication in PLOS One. Congratulations! Your manuscript is now being handed over to our production team.

Kind regards,

on behalf of

Dr. Anthony Mwinilanaa Tampah-Naah

Academic Editor

PLOS One